# A GDSL lipase-like from *Ipomoea batatas* catalyzes efficient production of 3,5-diCQA when expressed in *Pichia pastoris*

Sissi Miguel[1,6], Guillaume Legrand[2,6], Léonor Duriot[1], Marianne Delporte[2], Barbara Menin[3], Cindy Michel[1], Alexandre Olry[3], Gabrielle Chataigné[2], Aleksander Salwinski[1], Joakim Bygdell[4], Dominique Vercaigne[2], Gunnar Wingsle[5], Jean Louis Hilbert [2,7], Frédéric Bourgaud[1,7], Alain Hehn [3,7✉] & David Gagneul[2,7✉]

The synthesis of 3,5-dicaffeoylquinic acid (3,5-DiCQA) has attracted the interest of many researchers for more than 30 years. Recently, enzymes belonging to the BAHD acyl-transferase family were shown to mediate its synthesis, albeit with notably low efficiency. In this study, a new enzyme belonging to the GDSL lipase-like family was identified and proven to be able to transform chlorogenic acid (5-*O*-caffeoylquinic acid, 5-CQA, CGA) in 3,5-DiCQA with a conversion rate of more than 60%. The enzyme has been produced in different expression systems but has only been shown to be active when transiently synthesized in *Nicotiana benthamiana* or stably expressed in *Pichia pastoris*. The synthesis of the molecule could be performed in vitro but also by a bioconversion approach beginning from pure 5-CQA or from green coffee bean extract, thereby paving the road for producing it on an industrial scale.

[1] Plant Advanced Technologies, Vandœuvre-lès-Nancy, France. [2] UMR Transfrontalière BioEcoAgro N° 1158, Univ. Lille, INRAE, Univ. Liège, UPJV, ISA, Univ. Artois, Univ. Littoral Côte d'Opale, ICV – Institut Charles Viollette, 59000 Lille, France. [3] Université de Lorraine-INRAE, LAE, 54000 Nancy, France. [4] Chemistry Department, Umeå University, 90183 Umeå, Sweden. [5] Umeå Plant Science Centre, Department of Forest Genetics and Plant Physiology, Swedish University of Agricultural Sciences, 90183 Umeå, Sweden. [6] These authors contributed equally: Sissi Miguel, Guillaume Legrand. [7] These authors jointly supervised this work: Jean Louis Hilbert, Frédéric Bourgaud, Alain Hehn, David Gagneul. ✉email: Alain.Hehn@univ-lorraine.fr; David.Gagneul@univ-lille.fr

Plants can be seen as factories that produce a large variety of specialized molecules displaying interesting biological properties. Among them, phenolic compounds are reported for their human health benefits and especially for their antioxidant and anti-inflammation properties[1]. A well-known example is chlorogenic acid (5-O-caffeoylquinic acid, 5-CQA, CGA)[2–5] formed by the esterification of caffeic acid with quinic acid. This molecule can be further combined with a second caffeic acid to produce dicaffeoyl quinic acids like 3,4-dicaffeoylquinic acid or its isomer 3,5-dicaffeoylquinic acid (3,5-DiCQA)[2–5]. 5-CQA and 3,5-DiCQA were detected in many higher plants, and among them, the main sources are coffee, potato and vegetables[6–9]. According to epidemiological studies, their daily intake is between 100 and 224 mg for 5-CQA and between 0.2 and 7.7 mg for 3,5-DiCQA[6–8]. The increasing complexity of these caffeate derivatives provides additional properties to the molecules, such as a higher solubility or stability, but also new biological activities. Many reports highlight the health benefits of 5-CQA and 3,5-DiCQA[10,11]. For instance, 5-CQA prevents low-density lipid oxidation, whereas 3,5-DiCQA has antihepatotoxic and anti-inflammatory activities. Due to their neuroprotective activity largely described in the literature, 5-CQA and 3,5-DiCQA are considered for the treatment of Alzheimer's disease[12–16]. Additionally, 3,5-DiCQA and other chlorogenic acids have been associated to anti-virus-cell fusion properties against numerous human viruses like human immunodeficiency virus, influenza A, herpes simplex 1 and 2, Coxsackie B3, adenovirus and respiratory syncytial viruses[17–19]. In parallel to these anti-viral activities, 3,5-DiCQA has also been assigned as a potent anti-inflammatory compound against lipopolysaccharide-induced acute lung injury through the inhibition of human neutrophil elastase activity, making it a potential new therapeutic candidate against acute viral pneumonia diseases[20,21]. Finally, 3,5-DiCQA has also been identified as a hypotensive molecule, inhibiting angiotensin I-converting enzyme (ACE)[22,23]. This ACE-inhibiting activity could help protect the lung cells infected by SARS-CoV-2 since the virus is using angiotensin II-converting enzyme (ACE2), a protein homolog to ACE, as a cell entry point, which, in return, generates a disruption in the ACE/ACE2 balance[24].

The synthesis of 5-CQA is well documented, and four pathways have been described in the literature[2,4] (Fig. 1). In the first pathway, p-coumaroyl-CoA is esterified with quinic acid by a hydroxycinnamoyl-CoA: shikimate/quinate transferase (HCT), leading to p-coumaroylquinic acid[2–4]. This molecule is subsequently hydroxylated by a phenolic ester 3′ hydroxylase (C3′H) to generate 5-CQA[25]. A second pathway involves a hydroxycinnamoyl-CoA: quinate hydroxycinnamoyl transferase (HQT) using caffeoyl-CoA and quinic acid as substrates to produce 5-CQA[26]. This second route seems to be the main pathway in tomato[27]. A third route was reported in the late eighties involving hydroxycinnamoyl D-glucose as an alternative substrate for the acylation of quinic acid[28]. The enzyme, a hydroxycinnamoyl D-glucose: quinate hydroxycinnamoyl transferase, was purified from the root of sweet potato, but no further molecular clues are available at present. Recently, Barros and colleagues showed that an enzyme, a coumarate 3-hydroxylase (C3H), catalyzes the direct 3-hydroxylation of 4-coumarate to produce caffeate that is further converted by a 4-hydroxycinnamate: CoA ligase (4CL) in caffeoyl-CoA and fused to a quinic acid to produce 5-CQA[29]. Finally, caffeate can be produced from caffeoyl shikimate due to a caffeoyl shikimate esterase (CSE)[30].

The mechanism involved in the production of 3,5-DiCQA, although being widely investigated over the last 30 years, has not been fully elucidated. In 1987, Villegas et al. performed the first protein purification attempts from sweet potato tissues (Ipomoea batatas). The study performed by these researchers led to the characterization of the enzymatic properties of a chlorogenic acid:

chlorogenate caffeoyl transferase directly involved in the synthesis of this molecule using 5-CQA as the sole substrate[31]. The isolated enzyme consisted of a single polypeptide with a molecular mass of 25 kDa and a pI of 4.6. The gene encoding this enzyme has not been further investigated. More recently, several genes were characterized, and the corresponding enzymes were reported to catalyze the production of 3,5-DiCQA. Hence, an HCT isolated from coffee and an HQT from tomato were shown to be able to catalyze the esterification of 5-CQA in the presence of caffeoyl CoA or 5-CQA alone, respectively, when expressed in Escherichia coli[27,32]. HCT and HQT are members of the BAHD family of plant-specific acyl-CoA-dependent acyltransferases[33]. However, the catalytic efficiency monitored in vitro is insufficient to explain the high levels of 3,5-DiCQA present in several plant species. Additionally, the abovementioned BAHD transferases do not share the same biochemical properties as the enzyme previously isolated by Villegas[31]. It might therefore be possible that the biosynthesis of 3,5-DiCQA in I. batatas relies on a different mechanism that remains to be identified at the molecular level.

In this report, we carried out a combined proteomic and transcriptomic approach to identify and functionally characterize two enzymes involved in the final step of the synthesis of 3,5-DiCQA from 5-CQA in I. batatas: IbHCT (hydroxycinnamoyl-CoA: shikimate/quinate hydroxycinnamoyl transferase) belonging to the BAHD acyltransferase and IbICS (isochlorogenate synthase) to the GDSL lipase-like family. IbICS exhibits the highest efficient chlorogenic acid:chlorogenate caffeoyl transferase activity. A bioproduction process using Pichia pastoris was successfully established, and a valuable tool to produce 3,5-DiCQA at high levels in industrial applications was initiated.

## Results

**Identification and functional characterization of IbHCT.** BAHD acyltransferases have been previously shown to be involved in the synthesis of 3,5-DiCQA in Coffea canephora and Solanum lycopersicum. Therefore, our investigation first began by focusing on the identification of orthologous genes in I. batatas. Using the amino acid sequences of HCT from robusta coffee (ID: ABO47805) and tomato SlHQT (ID: Q70G32) as queries, a tBLASTn search was conducted on the GenBank Database. We focused on a putative IbHCT (protein ID: BAJ14794). This protein shares 85% amino acid identity with the HCT from C. canephora that was described to be involved in the synthesis of 3,5-DiCQA. This enzyme also shares 84% with an HCT from Nicotiana tabacum (ID: CAD47830) and 82% with that of Cynara cardunculus (ID: AAZ80046). The protein contains the HXXXD and DFGWG motifs characteristic of the acyltransferases of the BAHD family (Supplementary Fig. 1). Based on the sequence available, the coding sequence was amplified from RNA extracted from I. batatas roots. The resulting sequence shows six different amino acids compared to the registered sequence. Since these amino acid mismatches were reproducible through several cloning experiments and were localized in nonconserved regions, this sequence was used for the following experiments (Supplementary Fig. 1). Since an HQT has been described for the same activity in tomato, we also cloned a gene encoding a putative IbHQT (Protein ID: BAA87043) for which the sequence was already available in public databases.

To perform functional characterization of this enzyme, it was expressed in E. coli BL21 (DE3) strains and, after purification, incubated in the presence of caffeoyl-CoA and quinic acid at 30 °C and pH 7.0 for 1 h, as described in Comino et al.[3] (Fig. 2a). In the presence of IbHCT, a new product (P1), absent in control reactions, was detected (Fig. 2a.i). P1 was identified as 5-CQA on the basis of UV spectra, mass spectra (MS) and retention time in

**Fig. 1 Simplified biosynthetic pathway of 3,5-DiCQA. IbHCT and IbICS (in green) are two enzymes of *Ipomoea batatas* characterized in this study.** PAL L-phenylalanine ammonia-lyase, C4H cinnamate 4-hydroxylase, C3H 4-coumarate 3-hydroxylase, 4CL 4-coumaroyl-CoA ligase, HCT/HQT 4-hydroxycinnamoyl CoA: shikimate/quinate hydroxycinnamoyl transferase, C3'H 4-coumaroyl shikimate/quinate 3'-hydroxylase, CSE caffeoyl shikimate esterase, IbICS isochlorogenate synthase from *I. batatas*, IbHCT HCT from *I. batatas*.

comparison to an authentic standard (Fig. 2a.ii). P1 gave a $[M\text{-}H]^-$ ion at a mass-to-charge ratio ($m/z$) of 353 and exhibited a maximum absorption peak at 326 nm (Fig. 2a.iii). These results showed that IbHCT has the ability to synthesize 5-CQA as HCTs from other plant species[2–5]. An additional peak (not present in the initial reaction mixtures) corresponding to caffeic acid also appeared in the positive and control reactions. This molecule may arise from caffeoyl-CoA or 5-CQA degradation, as reported by Kojima and Kondo[34] and Lallemand et al.[32]. In addition to these results, incubation of recombinant IbHCT with 5-CQA and caffeoyl-CoA at pH 6.0 led to the production of two new compounds (P2 and P3; Fig. 2b.i) absent in the negative control after 1 h of incubation at 30 °C. P2 was identified as 3,5-DiCQA based on UV spectra (maximum absorption at 326 nm) and MS spectra ($[M\text{-}H]^-$ ion at a $m/z$ of 515) in comparison with an authentic standard (Fig. 2b.ii and iii). P3 eluted before 3,5-DiCQA

and was identified as 3,4-DiCQA by mass spectrometry (not shown). P3 may result from the isomerization of 3,5-DiCQA (Supplementary Fig. 11), as previously described in other studies[32]. These results demonstrated that IbHCT is able to catalyze both 5-CQA and 3,5-DiCQA synthesis in vitro. Regarding IbHQT, only a very slight activity towards the synthesis of p-coumaroylquinic acid was monitored. No production of DiCQA was detected under our experimental conditions.

HCTs are often described to be involved in the synthesis of various hydroxycinnamoyl esters. The specificity of the enzyme in in vitro conditions with cinnamoyl-, *p*-coumaroyl-, caffeoyl- or feruloyl-CoA as acyl donors in combination with quinate or shikimate as acceptors was assessed. As expected, in the presence of the enzyme cinnamoyl-quinate (P4 in Supplementary Fig. 2 A), *p*-coumaroyl-quinate (P5 in Supplementary Fig. 2B), cinnamoyl-shikimate (P6 in Supplementary Fig. 3A), *p*-coumaroyl-shikimate

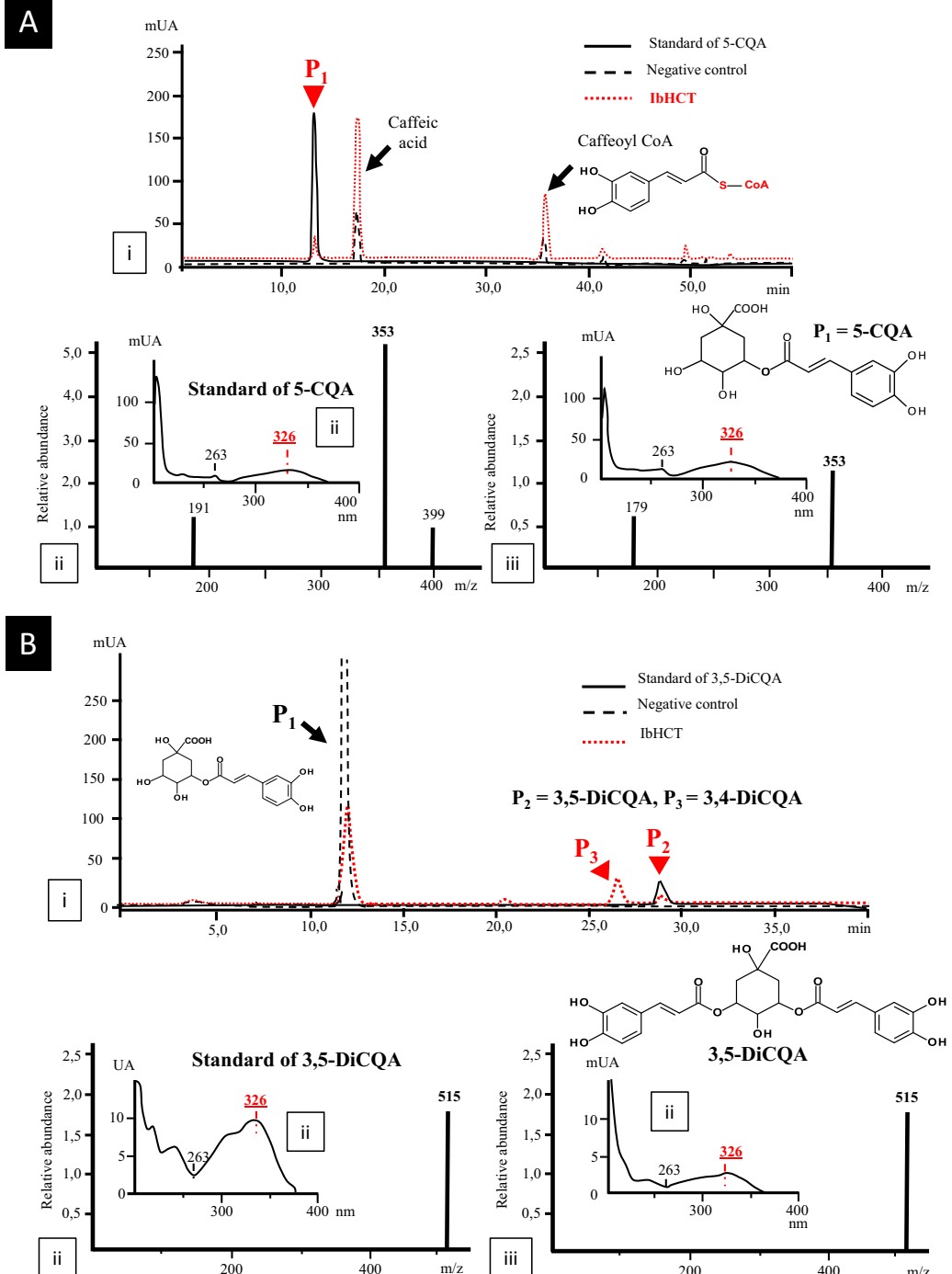

**Fig. 2 Enzymatic conversion of caffeoyl CoA and quinic acid in 5-CQA.** Synthesis of 5-CQA from caffeoyl CoA and quinic acid (**a**) and of 3,5-DiCQA from caffeoyl-CoA and 5-CQA (**b**) catalyzed by IbHCT after a 1 h incubation at 30 °C at pH 7.0 or 6.0, respectively. (i) HPLC profile of the reaction mixtures after incubation with the recombinant protein, of the standards or of reaction mixtures incubated in the absence of the recombinant protein; (ii) UV spectrum and MS profile of standards; (iii) UV spectrum and MS profile of metabolization products.

(P7, P8 and P9 in Supplementary Fig. 3B) and caffeoyl-shikimate (P10 in Supplementary Fig. 3C) were detected in the reaction mixtures containing the different combinations of substrates. However, derivatives with feruloyl entities were not detected in the presence of quinate or shikimate.

This study led us to three main conclusions. First, IbHCT has a broad substrate activity as orthologous enzymes from coffee and tobacco[2,32]. With the exception of the production of 3,5-DiCQA and its isomer, 3,4-DiCQA, when the enzyme was incubated with

caffeoyl-CoA and quinate, we could not identify any additional peak that could correspond to dicaffeoyl shikimic acid or any other dihydroxycinnamoyl acids. Second, in a preliminary experiment, the content of 3,5-DiCQA was higher in roots grown in hydroponic conditions without nitrogen (2 g of 3,5-DiCQA / 100 g dry weight) than in medium supplemented with nitrogen (1 g of 3,5-DiCQA / 100 g dry weight) (Supplementary Fig. 4). The results obtained regarding the low efficiency of IbHCT are insufficient to explain the high concentration of 3,5-DiCQA in

**Table 1 Analyses of the peptides obtained by trypsin digestion of the proteins collected by SDS-PAGE.**

| Acrylamide slice ID displaying an activity | Number of different peptides | Number of related proteins |
|---|---|---|
| 3 | 261 | 25 |
| 4 | 230 | 27 |
| 5 | 437 | 64 |
| 6 | 779 | 52 |

Slices of SDS-PAGE exhibiting 3,5-DiCQA synthesis activity were analyzed by LC MS/MS. Numbers of corresponding proteins identified in public databases are indicated.

*I. batatas* roots grown under nitrogen depletion conditions. Third, the molecular mass of the enzyme (50 kDa) does not match the value of the chlorogenic acid:chlorogenate caffeoyl transferase described previously by Villegas et al.[31]. Therefore, another strategy was initiated in an attempt to identify additional enzymes involved in the synthesis of 3,5-DiCQA.

**Identification and functional characterization of IbICS**

*cDNA library construction.* Little genetic information concerning *I. batatas* was available in public databases, which led us to generate a RNA-seq library. Total RNA was extracted from the roots of plants deprived of nitrogen. As mentioned above, in roots of plants grown under nitrogen-deprived conditions, the amounts of 3,5-DiCQA doubled compared to the control conditions. RNAs were extracted from roots of these plants to construct a 454-based de novo RNA-seq library (ID: PRJNA647243). This library was normalized to ensure representation of most of the genes expressed in these tissues. Clustering and assembly of these singletons led to the identification of 65264 contigs. These contigs were annotated and functionally assigned using Blast2GO. A Gene Ontology analysis realized on the library showed 8454 different annotations (Supplementary Fig. 5A), among which 37% were assigned to catalytic activities (Supplementary Fig. 5B). To assess the coverage of our library, we searched and confirmed the presence of IbHCT along with 13 additional sequences of putative acyltransferases belonging to the BAHD family. These enzymes were distributed in almost all BAHD acyltransferase clades (Supplementary Fig. 6).

*Preparation of IbICS enriched protein fractions and sequence analysis.* Based on the work performed by Villegas and collaborators, slices of *I. batatas* tubers were incubated in the dark for 48 h at 25 °C to induce 3,5-DiCQA accumulation and promote an increase in target enzyme activity[31]. Proteins from an acetone powder were extracted and subjected to different separation steps to identify the enzyme involved in 3,5-DiCQA biosynthesis. The enrichment procedure was only followed with the most active fractions after each step. In the final step, the proteins were separated by SDS-PAGE, and the proteins were collected in 8 different acrylamide gel fragments (~1 cm). The proteins extracted from each acrylamide piece were tested, and 3,5-DiCQA synthase activity could be highlighted in fragments #4 and #5, which correspond to proteins with an apparent molecular weight of approximately 30 kDa. The proteins present in these two bands, as well as the proteins included in the surrounding fragments (#3 and #6), were digested with trypsin and further analyzed using a UHPLC MS/MS approach. This analysis provided several hundred peptide sequences (Table 1, and Supplementary Table 1) that have been further identified and assigned to 25 (fragment #3) up to 64 (fragment #5) homologous proteins present in public databases.

*Identification of a full-length cDNA encoding IbICS.* The peptides identified by UHPLC MS/MS were used in a tBLASTn search of the de novo normalized RNA-seq library obtained from *I. batatas* roots to subsequently identify the corresponding coding sequences. A total of 24 proteins were identified in the four selected gel samples (Supplementary Table 1), and among these proteins, a sequence matching with a predicted *Ipomoea nil* GDSL esterase/lipase At1g28590-like (XM_019313071.1) was detected in fragments #3, #4 and #5. This result led us to identify IbICS, a 1158 bp coding sequence (CDS) (Genbank ID: MT291823) (Supplementary Fig. 7) encoding a putative 385-amino-acid-long protein (Supplementary Fig. 7 and Fig. 3) with a calculated molecular mass of 41.93 kDa and pI of 4.92. In silico analysis performed with the SignalP and TargetP programs revealed a predicted 25-amino acid long N-terminal signal that might target IbICS into the secretory pathway. Removing this sequence leads to a protein of 39.54 kDa with an unchanged pI. A GDSL motif was detected between amino acids 34 and 37 of the deduced IbICS amino acid sequence (Fig. 3). A sequence alignment with three additional characterized GDSL lipase-like enzymes demonstrated a notably high sequence identity. Within GDSL, IbICS belongs to the subclass of SGNH hydrolases that are characterized by the presence of four conserved amino acids in four conserved blocks (I, II, III and V). A phylogenetic analysis showed that these enzymes clustered with GDSL identified in *Rauvolfia serpentina* (AAW88230), *S. lycopersicum* (CBV37053.1) and *Alopecurus myosuroides* (CAG27610). All these enzymes were shown to be involved in specialized metabolism (Fig. 4)[35].

*Functional characterization of IbICS.* To determine the function of IbICS, the CDS was cloned into two different inducible expression systems based on *E. coli* and *Saccharomyces cerevisiae*. To analyze the effective expression of the protein in both systems, we added a polyhistidine tag at the C-terminus of the recombinant protein. The production of the protein was efficient in both systems (Fig. 5a, b). To further characterize the function, some enzymatic assays were performed in vitro in the presence of 5-CQA under the same conditions as previously described with the protein purified from the *I. batatas* tissues. Unfortunately, none of the proteins displayed any 3,5-DiCQA synthase activity under these conditions. We extended our investigation to a *Nicotiana benthamiana* plant expression system. The infiltrated leaves were harvested 96 h post infiltration, and the crude extract was used to perform incubations in the presence of 5-CQA at 28 °C for 22 h. UHPLC-MS analysis showed the appearance of a new product (P11) that has been identified as 3,5-DiCQA (molecular weight at 516) by mass spectrometry (mass in negative mode at 515) (Supplementary Fig. 8). This product was not detected when the crude extract was prepared from control leaves, which led us to assume that the enzyme is active in this system.

*Kinetic parameters and substrate specificity of the recombinant protein.* The protein expressed in *N. benthamiana* leaves was further purified using a His-Trap Ni column. The presence and identity of the recombinant protein were assessed by immunoblot analysis using an anti-His antibody (Fig. 5c). The activity of the purified recombinant protein was measured in vitro using 5-CQA as a substrate. The products of the reaction analyzed by UHPLC-MS confirmed the production of 3,5-DiCQA (Fig. 6a) with an m/z of 515 comparable to a commercial standard (Fig. 6c). No product was detected when 5-CQA was incubated in the presence of a protein extract of *N. benthamiana* infiltrated with an empty plasmid (Fig. 6b). The optimal pH and temperature were set at pH 6.3 and 39.9 °C, respectively (Supplementary Fig. 9). The $K_m$ was determined at 3.5 mM (Supplementary Fig. 9). The activity of IbICS with various substrates including acyl donors (various hydroxycinnamate

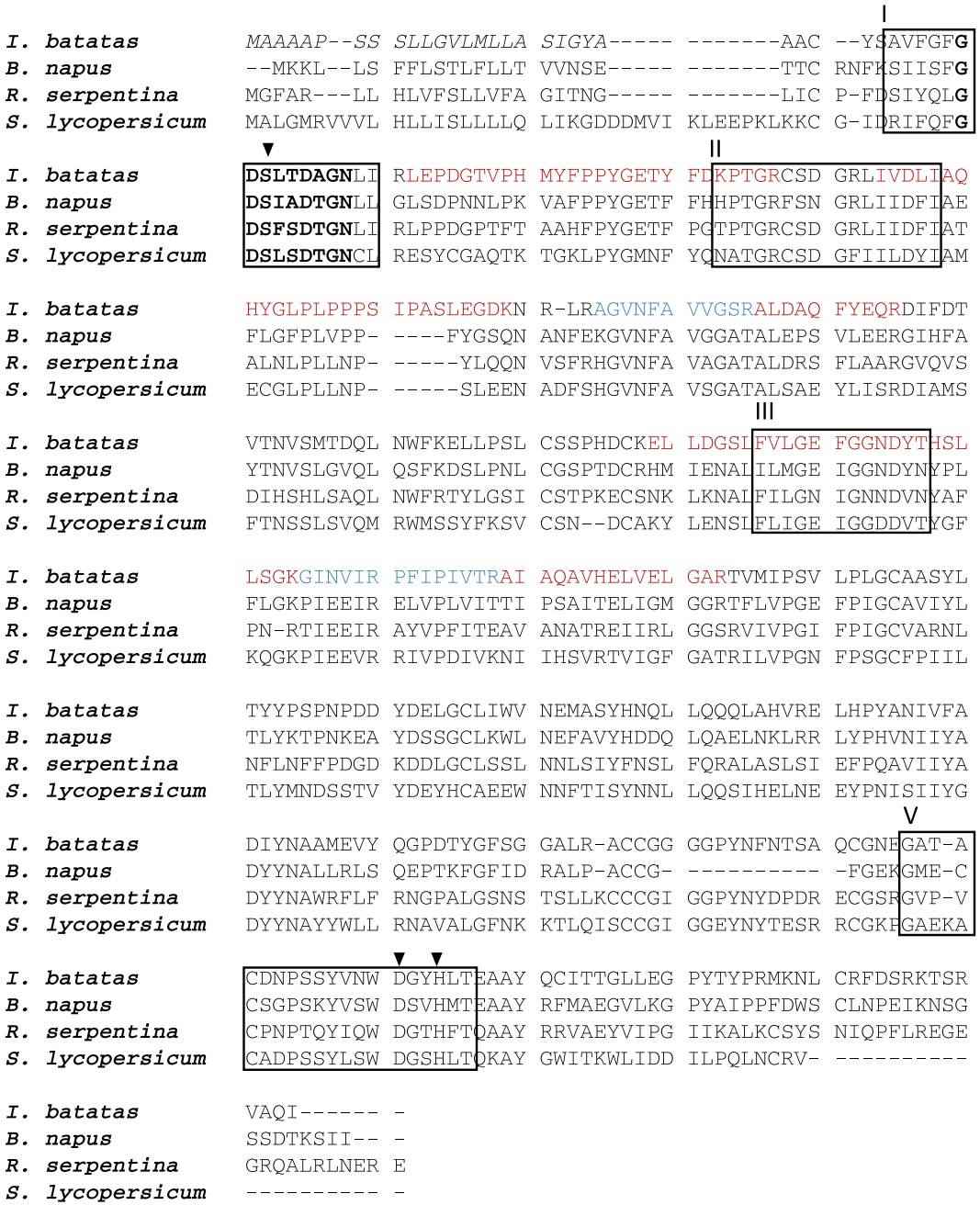

**Fig. 3 Alignment of the deduced amino acid sequence of IbICS with biochemically characterized GDSL lipase-like proteins.** The IbICS peptide sequence was deduced from cDNA and aligned with sinapine esterase from *Brassica napus* (AAX59709), acetylajmaline esterase from *Rauvolfia serpentina* (AY762990) and chlorogenate: glutarate caffeoyltransferase from *Solanum lycopersicum* (FR667689). The conserved GXSXXDXG motif is illustrated in bold. The predicted N-terminal leader sequences of all enzymes are shown in italics. Black box: conserved blocks in the SGNH-hydrolysase family (I, II, III and V). Amino acid residues forming the catalytic triad in the consensus sequences of blocks I and V are marked by black triangles. Sub-sequences identified by mass spectrometric sequencing after trypsin digestion are marked in red or in blue.

derivatives) and other acyl acceptors (5-CQA and quinic acid) was tested, but no activity was observed under our experimental conditions (Supplementary Fig. 10).

*Development of a system amenable to produce 3,5-DiCQA at industrial level.* To establish a suitable process for mass production of 3,5-DiCQA, we switched from the plant system to *P. pastoris* as an alternative expression factory. The IbICS CDS was cloned into the pICZαA integrative plasmid that contains the *S. cerevisiae* alpha secretor factor sequence favouring the excretion of the protein into the culture medium. The recombinant

*P. pastoris* strain was cultured under standard conditions, and protein expression was induced with methanol after 96 h. These conditions led to the production of the protein detectable in the culture medium (Fig. 5d). In a second step, different amounts of pure 5-CQA were added to the medium and the production of 3,5-DiCQA was monitored. The highest bioconversion rate of 5-CQA (61.3%) was obtained with 7.5 mM 5-CQA (Fig. 7a), leading to the synthesis of 2.3 mM of DiCQA (instead of 3.75 mM expected for 100% efficiency). Using a higher concentration of 5-CQA (over 10 mM) led to an inhibitory effect, and a lower concentration decreased the yields of 3,5-DiCQA produced

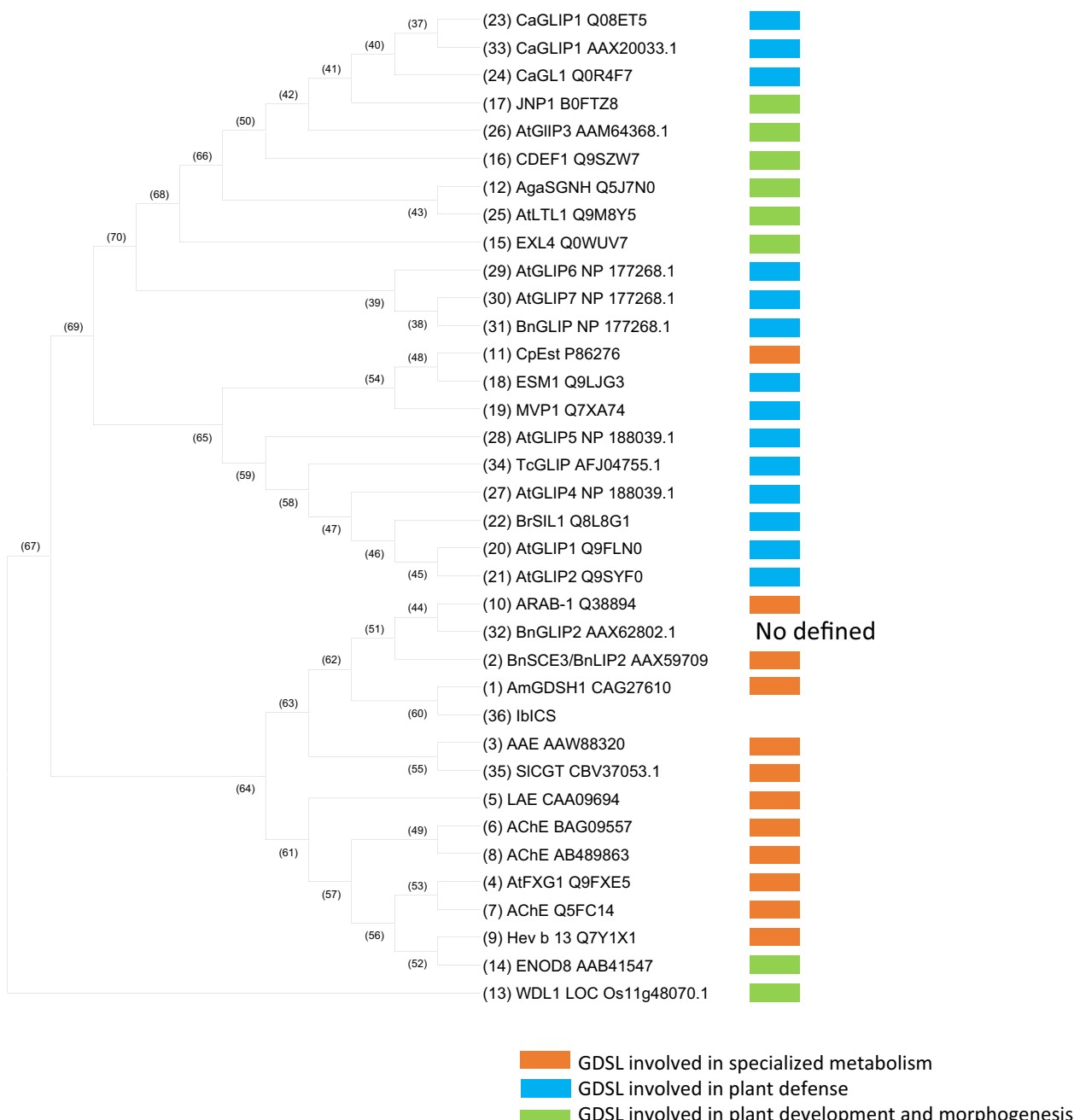

**Fig. 4 Phylogenetic tree of GDSL lipase/esterase protein sequences available in public databases.** The tree was constructed using the neighbour joining method and adapted from Chepyshko et al.[35].

(Fig. 7a). A 3,5-DiCQA-specific synthesis could also be highlighted when adding a green coffee extract containing 5-CQA (Fig. 7b). The bioconversion allowed a 7.5-fold increase of 3,5-DiCQA in comparison to native extract with a transformation efficiency estimated at 36%. The difference in efficiency in comparison with pure 5-CQA described above might be related to the complex composition of the green coffee bean extract where compounds structurally close to 5-CQA or other unrelated molecules might exert an inhibitory activity against IbICS.

## Discussion

Given the interest of 3,5-DiCQA, many scientists have been searching for more than 30 years for the gene(s) responsible for its biosynthesis. In this report, our phytochemical analyses show that a decrease in nitrogen availability increased the concentrations of 5-CQA acid and 3,5-DiCQA in *I. batatas* plant root tissues. The amount of 3,5-DiCQA can reach almost 2 g for 100 g of dried roots. Such results have already been described by Galieni and collaborators, who have evidenced in lettuce that the accumulation of phenolic acids was significantly enhanced, especially 5-CQA and 3,5-DiCQA[36]. Consequently, we carried out a combined proteomic and transcriptomic approach in this tissue to identify and functionally characterize two enzymes involved in the final step of the synthesis of 3,5-DiCQA from 5-CQA: IbHCT and IbICS.

In this study, an HCT isolated from *I. batatas* was shown to be able to synthesize 3,5-DiCQA using chlorogenic acid and caffeoyl CoA as substrates in addition to its main catalytic activity towards 5-CQA production. Such enzymatic activity has already been demonstrated for an HCT isolated from coffee. The synthesis of

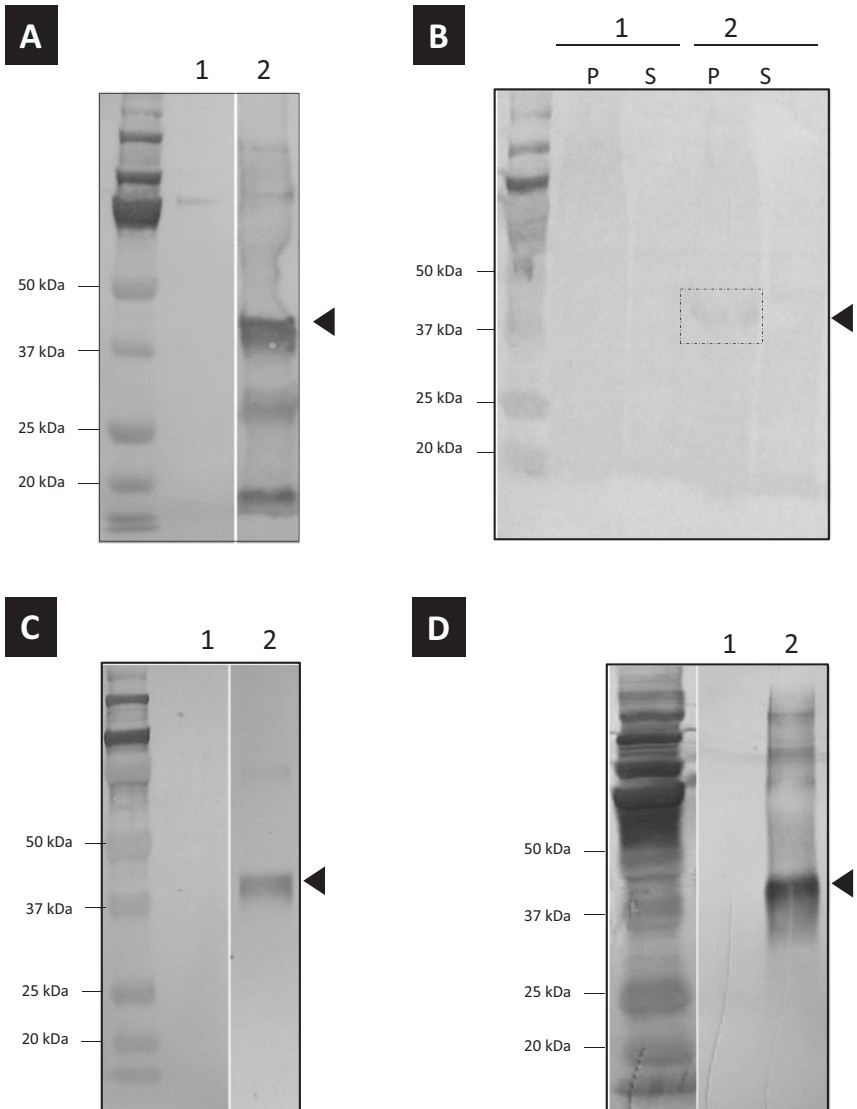

**Fig. 5 Expression analysis of IbICS in different heterologous expression systems.** Western blot analysis of proteins produced and purified from **a** *E. coli*, **b** *S. cerevisiae*, **c** *N. benthamiana* and **d** *P. pastoris*. For each sample, 1 corresponds to the purification performed on cells/tissues transformed with an empty vector. 2 corresponds to the purification performed on cells/tissues transformed with a plasmid containing the *IbICS* CDS. P: pellet of *S. cerevisiae* culture after protein extraction; S: supernatant. Parts of gels separated by a space were grouped together and lined up to facilitate the visualization of results.

3,5-DiCQA was found to be optimal at pH 6, while the synthesis of chlorogenic acid was optimal at pH 7 (not shown). This difference could be explained by the fact that the activity of the enzyme may change depending on the cell compartment. It is therefore likely that the activity of 3,5-DiCQA synthesis takes place in an acidic compartment of the cell, such as the vacuole[27]. We also observed that the reaction leading to the synthesis of 3,5-DiCQA was notably slow. Indeed, the product became detectable after a 1 h incubation time. These data are consistent with the results described by Lallemand et al. for coffee HCT[32]. Finally, small amounts of 3,5-DiCQA were produced in vitro, which did not allow quantification. According to several studies[2–4], HCTs perform reversible reactions. It is therefore possible that the low reaction yields are related to an equilibrium between the precursor and the product. However, the low catalytic performance of HCTs for the synthesis of 3,5-DiCQA makes it difficult to explain the accumulation of high concentrations of this molecule in plants like sweet potato. Some authors suggest that 5-CQA synthesized in the cytoplasm is further transferred into vacuoles, where it is stored as 3,5-DiCQA. This side activity

has been referred to as "moonlight activity" by Moglia and collaborators[37].

Using a proteomic/transcriptomic approach, a completely new player in the synthesis of these DiCQAs was isolated. *IbICS* is a gene encoding an enzyme belonging to a subgroup of the GDSL family called SGNH hydrolase. The name of this enzyme derives from the four strictly conserved Ser-Gly-Asn-His residues in four conserved blocks: I, II, III and V[38]. It has been reported that this enzyme is generally involved in the regulation of development, morphogenesis and the synthesis of specialized metabolites and compounds involved in defence mechanisms[35]. Our experiments showed that this enzyme could be the main actor in the synthesis of 3,5-DiCQA in sweet potato. Whereas the experiments performed with IbHCT led to the production of trace amounts of 3,5-DiCQA, we were able to convert nearly 80% of the 5-CQA that was added to the reaction mix containing IbICS in in vitro experiments. An enzyme using chlorogenate as an acyl donor (a chlorogenate-dependent caffeoyl transferase) was previously described in tomato roots, but no molecular evidence was provided[39]. In this case, the enzyme transfers the caffeoyl group of

5-CQA to ethanol to produce ethyl caffeate. This enzyme is inhibited by PMSF, which is a feature common to characterized members of the GDSL family and suggests that this enzyme could

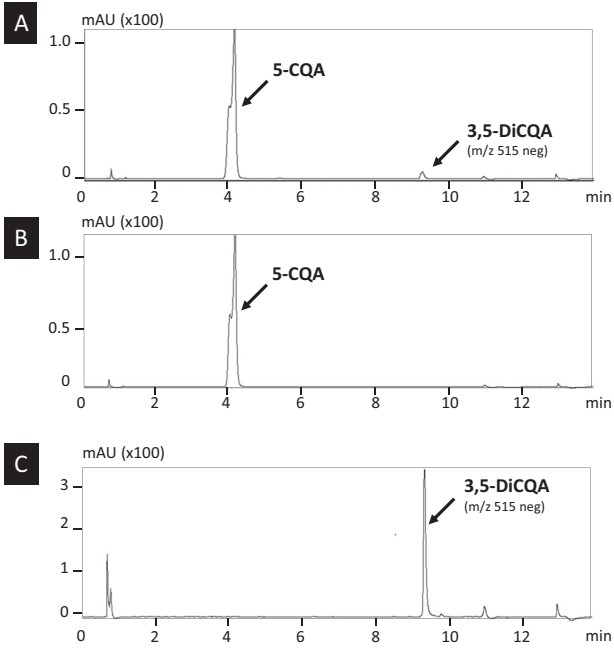

**Fig. 6 HPLC analyses of the in vitro metabolization product of 5-CQA. a** 3,5-DiCQA Commercial standard. **b** Incubation of 5-CQA with a protein mix prepared from leaves of *N. benthamiana* infiltrated with recombinant *A. tumefaciens* transformed with an empty plasmid. **c** Incubation of 5-CQA with purified IbICS-HIS produced in *N. benthamiana* leaves. Analyses were performed at 330 nm. The identity of 3,5-DiCQA was confirmed by MS.

also belong to the GDSL family[31,40,41]. This finding indicates an evolutionarily driven metabolic diversification that was already highlighted for SlCGT[42]. BAHD and SCPL acyltransferases are currently considered the two main families of acyltransferases involved in phenylpropanoid metabolism[40]. In the future, it might be relevant to add the GDSL family if additional examples of such activity are evidenced.

The expression of IbICS has been successfully realized in four heterologous expression systems. Under standard conditions, the expression in *S. cerevisiae* and *E. coli* was effective for both systems but no activity in 3,5-DiCQA synthase was detected. This finding could be related to a poor solubilization of the proteins or to post-translational modification. Regarding the second possibility, an in silico analysis using the online GlycoEP tool (http://crdd.osdd.net/raghava/glycoep/submit.html) revealed 3 putative N-glycosylation sites (N136, N319 and N334) and 7 putative O-glycosylation sites (T51, T69, T234, T320, T347, T355 and T356). The third expression system based on agroinfiltration of *N. benthamiana* leaves led to the production of a functional enzyme able to convert 5-CQA into 3,5-DiCQA. This result is consistent with other studies. For example, Tan et al. could not express a functional *Brassica napus* GDSL esterase (BnGLIP) in bacteria or yeast but succeeded in *N. benthamiana* leaves. In this system, these researchers obtained up to 50 μg of recombinant protein from 1 g of leaves[40]. Similarly, expression of SlCGT, a GDSL lipase from tomato, failed to be active in *E. coli* and yeast but was successfully produced and active in tobacco plants[42]. The plant expression system has also been used for the expression of CpLIP2, a GDSL lipase isolated from *Carica papaya*[41]. 3,5-DiCQA has been identified as a potential anti-viral and anti-inflammatory candidate molecule but it is scarcely available at industrial scale considering the low concentrations found in plants[43]. Hence, producing this molecule in high amounts is an interesting challenge. In our hands, the best method was the *P. pastoris* expression platform. The main advantages of this

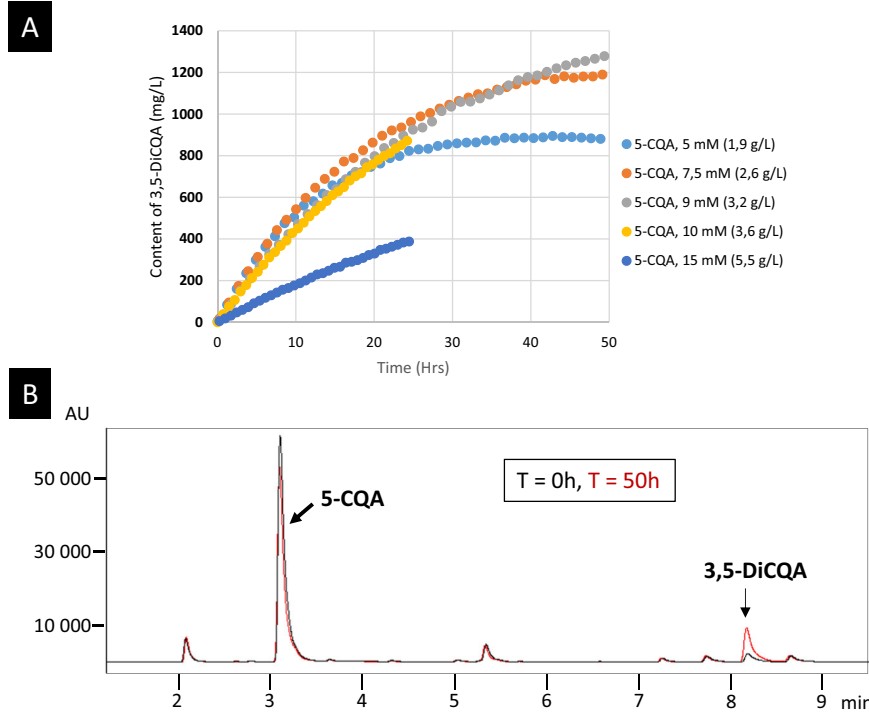

**Fig. 7 3,5-DiCQA synthesis using a bioconversion strategy with *P. pastoris* as a host platform. a** Time course of incubation of 3,5-DiCQA in the presence of different concentrations of pure 5-CQA. **b** Evaluation of the conversion efficiency in the presence of green coffee extract mix containing a high intrinsic amount of 5-CQA. Incubation was performed for 50 h. The analysis was performed at 330 nm.

system are that the protein is excreted in the culture medium and the IbICS produced is active. Other GDSL lipases were described to be active and are secreted out of the plant tissue. For example, JNP1, a GDSL lipase/esterase isolated from *Jacaranda mimosifolia*, a tropical tree, could be highlighted in the floral nectar[44]. GLIP1, another enzyme belonging to this same family, has been detected in the secretome of *Arabidopsis thaliana*. In this case, the enzyme is produced in response to salicylic acid treatments[45].

The proposed *Pichia*-based bioconversion system is rather efficient since 61.3% of the 5-CQA could be converted, leading to a final concentration of 1.2 g/L 3,5-DiCQA when starting from 2.6 g/L 5-CQA. This system could be further improved by using a fed batch approach consisting in adding substrate during the process without reaching substrate toxic concentration, which seems to be a limiting factor. Another possible process improvement might be to replace the pure substrate molecule and use a plant extract harbouring high concentration of 5-CQA. This last strategy was tested in our experiments but the conversion rate of the 5-CQA present in a green coffee extract could only reach 36%. This result might be related to the presence of numerous specialized metabolites that can display an inhibitory effect on the GDSL enzyme. Finally, IbICS is soluble and active in aqueous solution. However, in our complex culture medium, with an improper pH and temperature, there might be a decline in the enzymatic activity, despite preservative buffers. The production yield might, for example, be increased through enzyme immobilization approaches that would preserve it from degradation. Such approaches were investigated in several cases and were described to provide interesting benefits, such as convenient handling, easy and efficient recovery of the product and reuse of the same enzyme batch in several reactions of conversion[46,47]. This finding was reported for a lipase isolated from *Candida rugosa* and immobilized on magnetic hollow mesoporous silica microspheres. In this case, the enzyme was used with fatty acids and triglycerides in a solvent-free system. The authors obtained a conversion rate of 90%, and the enzyme could be reused in 50 successive cycles.

**Other potential players involved in 3,5-DiCQA biosynthesis**. Our results also raise new questions. The investigation of the biochemical properties of IbICS revealed some discrepancies with the pioneering work of Villegas et al.[31]. First, during the purification process, ICS activity was detected in protein fractions precipitated with 40% ammonium sulphate, while in a previous report, the enzyme precipitated between 40 and 70% saturation. Second, IbICS has a calculated molecular weight of 39.54 kDa and a pI of 4.92, whereas the chlorogenic acid:chlorogenate caffeoyl transferase isolated by Villegas et al. had a deduced molecular weight of 25 kDa and a pI of 4.6. Third, the kinetic parameters are highly different. We found a $K_m$ of 3.5 mM and an optimal pH of 6.3, while Villegas determined a $K_m$ of 0.87 mM and an optimal pH of 5.0, with the enzyme losing 50% of its activity at pH 6.8. Taken together, these differences between the two studies suggest that additional enzymes might be involved in the outstanding efficient synthesis of 3,5-DiCQA in sweet potato roots. In our study, we overcome a new step thanks to the identification of IbICS. However, given the complexity of specialized metabolism, it is possible that we have not reached the end of the story. Further experiments are needed to test new hypotheses, such as the deletion of *IbICS* in sweet potato, through genome editing.

## Methods

**Plant material**. *I. batatas* were grown in hydroponic conditions (Supplementary Fig. 4). Young cuttings of sweet potato roots were soaked in water for 2 weeks to induce root generation. During the following 3 weeks, the plants were cultivated either in nitrogen-enriched medium (15:10:30, N:P:K, PlantProd) or in nitrogen-free medium (0:15:40, PlantProd) with electroconductivity adjusted each day to 1.0–1.2 and a pH of 5.6–5.8.

For enzyme purification, sweet potato tubers were purchased at a local supermarket. Discs (1 cm diameter) were prepared from tuber slices (5 mm thick) using a cork borer, rinsed with distilled water and immediately placed on moist filter paper in closed plastic boxes. Discs were incubated at 25 °C in the dark for 48 h and subsequently used as a source for identifying the enzyme.

*N. benthamiana* seeds were sown on soil and cultivated in culture rooms under a 16 h/8 h day/night photoperiod with artificial light (70 μmol m$^{-2}$ s$^{-1}$) at 26 °C and 70% humidity. Three-week-old plantlets were transplanted into individual pots and were used after 3–4 weeks for agroinfiltration experiments, as described in Munakata et al.[48].

**Cloning, expression in *E. coli* and purification of IbHCT**. To amplify the *IbHCT* coding sequence (ID: BAJ14794), we used IbHCT_Fw: 5′-CGGTTTGGATC-CAATGAAGATCAGCGTGAAGGAC-3′ and IbHCT_Rev: 5′-CGCTCGAGAA-TATCATACAGGAACTCTTTGAAG-3′ primers. First-strand cDNAs were synthesized from RNA extracted from *I. batatas* roots cultivated in nitrogen-free medium and using the SuperScript™ III One-Step RT-PCR System with Platinum™ *Taq* High Fidelity DNA Polymerase (Invitrogen™, Thermo Fisher Scientific). The PCR products were ligated into the pCR8 plasmid (Invitrogen™, Thermo Fisher Scientific) and further transferred between the *Nde*I and *Xho*I restriction sites in the pET28b expression vector (Addgene, https://www.addgene.org/). Expression in *E. coli* and purification of the proteins were performed as described by Comino et al.[3].

**Enzymatic in vitro assays of IbHCT**. Synthesis of 5-CQA, cinnamoyl-shikimate, *p*-coumaroyl-shikimate and caffeoyl-shikimate from cinnamoyl-, *p*-coumaroyl- and caffeoyl-CoA, respectively, with quinic or shikimic acid was performed in a volume of 400 μL with 0.6 mM of ester-CoA (p-coumaroyl-, cinnamoyl-, caffeoyl-CoA), 0.6 mM of quinic acid (or shikimic acid) and 15 mg/mL of purified enzyme in 0.1 M NaPi at pH 7. The reactions were incubated at 30 °C during 1 h. The reactions were stopped by adding of 10 μL of mix acetonitrile/HCl 2 M and the solution was mixed and centrifuged at 10,000 × *g* for 10 min at room temperature. Products were then purified by ethyl acetate extraction for HPLC analyses.

Synthesis of 3,5-DiCQA from caffeoyl-CoA and 5-CQA was performed in a volume of 200 μl with 0.1 mM of caffeoyl-CoA, 0.1 mM of 5-CQA and 15 mg/ml of purified enzyme in 0.1 M NaPi at pH6. The reaction was incubated at room temperature during 15 h, stopped by adding of 5 μL of mix acetonitrile/HCl 2 M, mixed and centrifuged at 10,000 × *g* for 10 min at room temperature. Product was directly analyzed in HPLC and MS.

**Construction of an RNA-seq library**. Total messenger RNA (mRNA) was extracted from *I. batatas* roots cultivated in nitrogen-free medium using the Plant RNeasy® kit (Qiagen). A de novo normalized RNA-seq library was constructed using GS FLX + chemistry by Eurofins MWG Operon (Ebersberg, Germany). Deep sequencing of 600,000 ESTs was performed. Clustering and assembly of the reads were performed by the supplier.

**Preparation of the IbICS-enriched protein fraction and monitoring of ICS activity**. Sweet potato tuber discs (200 g) were chopped with a Waring Blendor in the presence of precooled acetone (−20 °C) (4 times 30 s runs with 30 s cooling-off periods). The homogenate was filtered through filter paper (Whatman), and the residue was washed with cold acetone (−20 °C) until the flow-through became transparent. After drying at room temperature, the acetone powder was homogenized in extraction buffer (50 mM Hepes-KOH, 10% glycerol (v/v), 2 mM EDTA, 10 mM MgCl₂, 1 mM DTT, 1 mM PMSF, 10% PVPP (m/v), pH 7.5). The suspension was further filtered on Miracloth and centrifuged at 12,000×*g* at 4 °C for 20 min. The proteins present in the supernatant were precipitated using ammonium sulphate (40% saturation) before being suspended in buffer A (50 mM Tris-HCl, pH 7.0). After centrifugation to remove insoluble residues (4000 × *g*, 5 min, 4 °C), the extract was loaded on a Sephadex G-50 column equilibrated with buffer A, and the proteins eluted with buffer A (0.4 mL min$^{-1}$). One-millilitre fractions were collected. The fractions containing enzyme activity were pooled and concentrated using a Vivaspin 6 centrifugal concentrator (100,000 MWCO, Sartorius). The protein solution was applied to a DEAE Sepharose anion exchange column equilibrated with buffer A and eluted with a step gradient of NaCl (from 0.15 to 1 M) prepared in 50 mM Tris-HCl, pH 7.0, at a flow rate of 1 mL min$^{-1}$. Active fractions were pooled and concentrated through a Vivaspin 6 centrifugal concentrator. SDS-PAGE analysis was conducted as described elsewhere[49]. After electrophoresis, one lane of the gel was stained with Coomassie blue, while another lane was washed twice with 2% Triton (v/v) to remove SDS, which might interfere with the enzyme activity. To localize enzyme activity, the two lanes were identically and separately cut into 8 slices (between 20 and 75 kDa, 1 cm height). The slices of the lane prewashed with Triton were ground, buffer A was added to solubilize the proteins, and thereafter, enzyme activity was measured in each preparation. Slices containing activities (4 slices) were identified, and the corresponding slices of the lane stained with Coomassie blue were subjected to LC/MS/MS analysis.

For the standard assay of IbICS activity, enzyme solutions were incubated at 30 °C for 15 min in 35 mM acetate buffer, pH 5.0, containing 1 mM 5-CQA and 10 mM MgCl$_2$ in a total volume of 100 μL. Reactions were stopped by placing the tubes in boiling water for 10 min. After centrifugation at 12,000 × $g$ for 3 min at 4 °C and filtration (0.45 μm filter), the reaction mixtures were analyzed by HPLC to quantify 3,5-DiCQA.

**Protein MS analyses.** The proteins collected by SDS-PAGE were digested in the gel in the presence of 50 mM ammonium bicarbonate, 0.01% Protease Max (Promega) and 12 ng μL$^{-1}$ of sequencing grade trypsin (Promega). After 2 h at 37 °C, the liquid phase of the digestion was acidified with trifluoroacetic acid (final concentration: 0.5% (v/v)). Samples were cleaned with an off-line C18 column before analysis by mass spectrometry.

The peptides were analyzed by ESI-LC/MS/MS using a nano LC system (Waters) coupled to a SynaptG2 mass spectrometer. Separation was performed using an Acquity BEH C18 nano reverse phase (150 cm) column (Waters) at a flow rate of 300 nL min$^{-1}$ with a mobile phase that consisted of solvent A (0.1% formic acid in water) and solvent B (0.1% formic acid in acetonitrile). Chromatographic separation was achieved using a 180 min linear gradient from 0 to 50% solvent B. Processing, deconvolution and peptide detection were performed using Data Analysis software (Protein Lynx global server, Waters).

**Bioinformatic analysis.** The FASTA file provided by MWG Eurofins Operon (Ebersberg, Germany) was imported as a library into BioEdit software (http://www.mbio.ncsu.edu/bioedit/bioedit.html). The peptide sequences obtained by MS analysis were compared with the RNA data available in the cDNA library using the Basic Local Alignment Search Tool (tBLASTn) to identify the corresponding coding sequences. A global analysis of the RNA-seq database was performed with the Blast2GO platform.

The prediction of putative N-terminal signal peptide (SP) was performed using SignalP 4.1 (http://www.cbs.dtu.dk/services/SignalP/) and TargetP 2.0 (http://www.cbs.dtu.dk/services/TargetP/). The molecular weight and isoelectric point (pI) of the protein and the number of amino acids were calculated using the ExPASy compute pI/MW tool (http://web.expasy.org/compute_pi/). To predict putative amino acid sites for glycosylation, we used the NetNGlyc 1.0 and NetOGlyc 4.0 servers (http://www.cbs.dtu.dk/services/NetNGlyc/ and http://www.cbs.dtu.dk/services/NetOGlyc/) at the Center for Biological Sequence Analysis website (http://www.cbs.dtu.dk/services/), as well as the GlycoEP tool (http://crdd.osdd.net/raghava/glycoep/submit.html).

The phylogenetic analyses were performed using Clustal W[50]. Pairwise distance matrices were generated using MEGA X[51]. Phylogenetic trees were reconstructed using the neighbour-joining algorithm from MEGA X.

**Cloning of *IbICS* for heterologous expression in *E. coli*, *S. cerevisiae*, *N. benthamiana* and *P. pastoris*.** To amplify the *IbICS* coding sequence (CDS), we used IbICS_Fw: 5′-ATGGCCGCCGCAGCTCCTTCTTCCTCGTTGCTTGG-3′; IbICS_Rev: 5′- TTAGTGGTGGTGGTGGTGGTGAATTTGGGCAACTCGAGATGTTTTGCGA-3′ primers. The reverse primer included the addition of a 6xHis tag at the C-terminal end of the protein (underlined in the sequence). The CDS was amplified from total RNA using the SuperScript™ III One-Step RT-PCR System with Platinum™ *Taq* High Fidelity DNA Polymerase (Invitrogen™, Thermo Fisher Scientific). The PCR product was ligated into the TA cloning vector pCR™8/GW/TOPO™ system (Invitrogen™, Thermo Fisher Scientific) and subsequently sequenced. The gene was further cloned by LR recombination using LR Clonase II™ (Invitrogen™, Thermo Fisher Scientific) into pEAQ-HT-DEST1[52] and pYeDP60GW[53] for expression in *N. benthamiana* and *S. cerevisiae*, respectively. The gene without signal peptide was amplified thanks to IbICS_Rev associated with IbICS_Ecoli_Fw: 5′-ATGTGCTATTCGGCCGTTTTCGGC-3′ from pEAQ-HT-IbICS, ligated into pCR™8/GW/TOPO™ system and cloned by LR recombination into p0GWA[54] for expression in *E. coli*.

To express IbICS in *P. pastoris* and direct the protein in the extracellular space, the putative N-terminal peptide signal of IbICS was replaced by the α-factor secretion signal of *S. cerevisiae*. The CDS was amplified with IbICS_Pichia_Fw: 5′-GAATTCTGCTATTCGGCCGTTTTCGGCCTTC-3′ and IbICS_Pichia_Rev: 5′-TCTAGAAAAATTTGGGCAACTCGAGATGTTTTGCGAG-3′ primers. Additional *Eco*RI and *Xba*I restriction sites were integrated (underlined nucleotides) to allow insertion into the pPICZαA vector (Invitrogen™, Thermo Fisher Scientific).

**Expression, purification and in vitro assays of recombinant IbICS produced in *N. benthamiana*.** The recombinant pEAQ-HT-IbICS plasmid (and the negative control corresponding to pEAQ-HT without the gene of interest) was introduced into *Agrobacterium tumefaciens* EHA105 using the freeze-thaw method[55]. The recombinant bacteria were selected on YEB medium (10 g L$^{-1}$ beef extract, 5 g L$^{-1}$ yeast extract, 10 g L$^{-1}$ peptone, 15 g L$^{-1}$ sucrose, 0.5 g L$^{-1}$, MgSO$_4$, 20 g L$^{-1}$ at pH 7.2) supplemented with 100 mg L$^{-1}$ rifampicin and 30 mg L$^{-1}$ kanamycin at 28 °C for 3 days.

For transient expression in *N. benthamiana*, recombinant agrobacteria were cultured in YEB medium containing 100 mg. L$^{-1}$ rifampicin and 30 mg. L$^{-1}$

kanamycin at 28 °C for 2 days at 180 rpm. Four hours before infiltration, acetosyringone was added to the cultures at a final concentration of 100 μM. The agrobacteria were centrifuged for 15 min at 5000 × $g$, and the pellet was washed twice with YEB medium to remove antibiotics. The pellet was finally suspended in infiltration buffer (10 mM MES, pH 5.6) at a cell density corresponding to OD$_{600}$ 0.8 ± 0.1. The whole aerial parts of *N. benthamiana* plants were immersed in the agrobacteria suspension and subjected to vacuum infiltration at 15 mbar for 20 min. Finally, the infiltrated plants were cultivated under a 16 h/8 h day/night photoperiod at 26 °C and 70% humidity. Leaves were harvested 6 days post infiltration and stored at −80 °C until use.

Total protein extract was obtained by grinding the leaves in the presence of extraction buffer (20 mM sodium phosphate, 0.5 M NaCl, pH 7.4) using a Polytron® PT2500E (Kinematica). The crude plant extract was clarified by centrifugation at 14,000 × $g$ for 10 min at 4 °C. The supernatant was further filtered through a 0.2 μm filter and loaded onto 1 mL HisTrap HP columns (GE Healthcare). Due to elution with 500 mM imidazole, one-millilitre fractions were collected and individually analyzed by SDS-PAGE. The fractions harbouring the target protein were pooled, diafiltrated with 5 volumes of phosphate buffer and concentrated using centrifugal 10 kDa molecular mass cutoff filters (Amicon® Ultra 0.5 mL Centrifugal Filters — PMNL 10 kDa, MILLIPORE). Purified proteins were stored at −20 °C.

The standard enzymatic in vitro assay was performed in a volume of 50 μL with 0.2–0.4 μg of purified enzyme. For evaluation of the optimal pH, the reaction mixture was incubated for 30 min at 25 °C in a buffer (0.1 M acetic acid, 0.1 M Tris, 20 mM MES) at pH values ranging from 4.0 to 9.0 with a single 5-CQA concentration of 10 mM. To evaluate the best substrate concentration, the 5-CQA was modulated from 0 to 30 mM. The reactions were stopped by adding 150 μL absolute ethanol, and the solution was mixed and centrifuged at 14,000 × $g$ for 10 min at 4 °C. The supernatant was analyzed by UHPLC-MS. Reactions were run in triplicate. $K_m$ value was determined in triplicate by fitting Michaelis–Menten curves using SigmaPlot Software.

**Expression of IbICS in *E. coli* and *S. cerevisiae*.** Expression in *E. coli* was realized in the BL21 Rosetta strain, and the culture was performed as for CcHCT described in Comino et al.[3]. Expression in *S. cerevisiae* was performed in the WAT11 yeast strain[56], as described for cytochrome P450s[57].

**Expression and in vivo bioconversion of 5-CQA to 3,5-DiCQA by recombinant IbICS produced by *P. pastoris*.** The *P. pastoris* X-33 strain (Invitrogen™, Thermo Fisher Scientific) was transformed as specified by the supplier with pPICZαA-IbICS or empty vector pPICZαA used as a negative control. The culture of recombinant *P. pastoris* was performed as described by the supplier. Briefly, a colony was picked into 25 ml of culture medium including glycerol at 28 °C at 250 rpm for 16–18 h. The yeast was centrifuged at 5000 × $g$, and the pellet was suspended in 250 mL of culture medium buffered at pH 6.0 and containing 0.5% methanol as an inducer. The induction was maintained for 3 days and supplied with methanol every 24 h. Three days after the onset of induction, 10 mM 5-CQA (final concentration) was added directly to the culture medium, and 3 days later, the supernatant of *Pichia* culture was sampled and analyzed by UHPLC-MS.

**Determination of protein concentrations and immunoblot analysis.** Protein concentrations of the extracts were determined using the Qubit™ Fluorometer (Invitrogen™, Thermo Fisher Scientific) with the Qubit™ protein assays kit (Invitrogen™, Thermo Fisher Scientific) according to the supplier's instructions. For immunodetection, 25 μL of protein mix was analyzed by SDS-PAGE. The separated proteins were blotted on a polyvinylidene fluoride membrane (Membrane PVDF 0.45 μm Amersham™ Hybond™ P, GE Healthcare). His-tagged IbICS was detected using a rabbit primary polyclonal anti-6xHistidine Epitope Tag (Cat#R1181, Acris) at a 1:5000 dilution in PBS (137 mM NaCl, 2.7 mM KCl, 8.1 mM Na$_2$HPO$_4$, 1.5 mM KH$_2$PO$_4$, pH 7.4) and a secondary anti-rabbit antibody conjugated to alkaline phosphatase activity (Cat#A0418−1ML, Anti-Rabbit IgG (whole molecule)–Alkaline Phosphatase antibody produced in goat, Sigma-Aldrich) diluted to 1:6000 in PBS. Revelation was performed with NBT/BCIP (Promega, USA) as substrates.

**Chromatography analyses.** Quantification of 5-CQA and 3,5-DiCQA was performed by:

a. UHPLC Shimadzu Prominence XR system (Shimadzu, Kyoto, Japan) with a PDA detector (applied range: 190–800 nm, detection at 300 nm) using a LiChrospher® 100 RP-18 reverse phase column (250 mm×4.6 mm, 5 μm, Merck, Darmstadt, Germany), maintained at 30 °C during all analyses. The mobile phase was composed of water, pH 2.6 adjusted using orthophosphoric acid (A) and pure methanol (B), delivered at 0.7 mL/min with the gradient of B phase as follows: 23–55% (0–43 min); 55–70% (43–50 min); 70% (50–55 min), 70–23% (55–60 min), 23% (60–65 min);

b. UHPLC Shimadzu Nexera X2 system (Shimadzu, Kyoto, Japan) with a PDA detector (applied range: 220–370 nm, detection at 300 nm) coupled to a mass spectrometer LCMS2020 (electrospray ionization in a negative ion mode, $m/z$

100–1000) using a Kinetex Biphenyl reverse phase column (150 mm × 2.1 mm, 2.6 μm, Phenomenex, Torrance, CA, USA), maintained at 40 °C during all analyses. The mobile phase was composed of water containing 0.1% formic acid (A) and pure acetonitrile (B), delivered at 0.5 ml/min with a gradient of B phase as follows: 5–25% (0–10 min); 25–90% (10–10.5 min) 90% (10.5–12 min), 90–5% (12–12.1 min), and 5% (12.1–14.1 min).

The contents of 3,5-DiCQA and 5-CQA were determined using the standards of 3,5-DiCQA and 5-CGA, respectively (both 100 mg/L in DMSO/water, pH 3.0). The concentrations of other hydroxycinnamate derivatives were expressed as 3,5-DiCQA equivalents.

## Data availability

The data that support the findings of this study have been deposited in Genbank (https://www.ncbi.nlm.nih.gov). The 454-based de novo RNA-seq library has the ID: PRJNA647243. The IbICS CDS has the ID: MT291823.

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

## Acknowledgements

The authors would like to thank Clément Charles for his technical support. This work was financially supported by the "Bioprolor2" project (Région Grand-Est) (to A.H.) and by the "Impact Biomolecules" project of the "Lorraine Université d'Excellence" (Investissements d'avenir–ANR) (to A.H.). Experiments performed at Charles Viollette Research Institute were supported by an ALIBIOTECH program (2015–2020) obtained from the CPER/FEDER Région Hauts-de-France. Marianne Delporte and Guillaume Legrand were supported by doctoral fellowships from the doctoral school 104 SMRE.

## Author contributions

M.S. performed the genetic molecular cloning, the bioinformatic analysis and the enzymatic assays in the 4 expression systems for IbICS. L.G., D.M., C.G., V.D., J.L.H. and G.D. purified the protein from the roots of *I. batatas*. B.J. and W.G. performed the MS protein analyses. L.D. cloned, expressed the IbHCT and realized the functional characterization. M.B. constructed the *I. batatas* cDNA library. O.A., S.A. and M.S. performed the enzymatic characterization of IbICS. M.S. realized the experimentations with green coffee extracts and S.A. performed the UHPLC-MS investigations of IbICS. M.S., F.B., H.A., and G.D. wrote the article. H.J.L., B.F., H.A. and G.D. supervised the project and are the principal investigators.

## Competing interests

The authors declare no competing interests.
