## [Peer Review File · Communications Biology]

Reviewers' comments:

Reviewer #1 (Remarks to the Author):

This manuscript reports a novel enzyme, IbICS, responsible for the production of dicaffeoylquinic acid in *I. batatas*. The authors also demonstrated a biotechnological production of the compound using pure monocaffeoylquinic acid and green coffee bean extract. Researchers have been long trying to identify this enzyme but have not succeeded. Therefore, the discovery of this enzyme is a breakthrough in natural product biosynthesis which is worth to publish. Although gene knockout experiment has not been performed, it is difficult to conclude as the author already suggested that IbICS is the main enzyme responsible for the biosynthesis. Additional experiments can be performed to support the finding. Please find my comments below.

Although English in the manuscript is understandable, it can be further improved.

Some parts of results are mixed with discussion. Please check, e.g. line 302-303 and 389-390.

Line 63-64, please provide a reference for daily intake of CQA and diCQA.

Figure 1, in the pathway leading to 3,5-DiCQA, quinate should be represented as one of the products catalyzed by IbICS.

Since HCT and HQT from some plants have been reported to be (partially) responsible for diCQA production, this manuscript only focused on HCT but did not raise the possibility of IbHQT in the diCQA production. I would like to see a discussion on why HQT is not mentioned/focused. Also in term of gene expression analysis, I did not see any results on IbHCT and IbICS. Is the expression pattern of IbICS correlated with diCQA accumulation in different parts/tissues of sweet potato? How were the expressions of those genes under nitrogen depletion experiments?

Please add more details on CQA profiles in sweet potato used in this study. Different genotypes can accumulate different profiles of CQA. Besides 3,5-diCQA as I understand it is the major diCQA in sweet potato, *I. batatas* also accumulate different types of diCQA. I wonder if the biosynthesis of other diCQAs is derived from different enzymes or isomerization via non-enzymatic reaction. This can be discussed. I also like to see the MS2 spectrum of the product to confirm that it is 3,5-diCQA. Depending on chromatographic condition, two diCQA isomers can be eluted at almost the same retention time.

I could not find M&M on IbHCT activity assay demonstrated in Fig 2A. Fig 2A(i), the authors mentioned that they detected increased caffeic acid content in both positive and control reactions that possibly derived from degradation of caffeoyl-CoA or 5-CQA. I wonder why the increased caffeoyl-CoA content can also be observed.

Fig. 4, please use color that clearly show the different between specialized metabolism and development/morphogenesis. What does "?" represent? Should you use IbICS in the figure?

Figure 5, the authors did not mention the amounts of proteins loaded into each lane. So how could the production level in different host be compared?

Could the reason for having no enzymatic activity be that the authors did not remove signal peptide when expression the recombinant protein in *E. coli*? Signal sequence is usually removed when expressing in prokaryotic system. And in *S. cerevisiae*, the recombinant protein could not be solubilised, therefore, no activity can be detected.

Figure 5C, the authors did not show purity of the purified enzyme in SDS-PAGE, only western blot result was provided. So I do not know the purity of the enzyme when determining kinetic

parameters.

Line 310, Galieni et al. did the experiment on lettuce? Not yarrow.

Have the RNAseq raw data been deposited?

Reviewer #2 (Remarks to the Author):

The reviewed manuscript deals with the identification of a new enzyme belonging to the GDSL lipase-like family that is involved in the final stage of transformation of 5-CQA into 3,5-diCQA and able to increase the transformation efficiency by over 60%, making the transformation process a valuable technological tool that can be easily transferred on an industrial scale. To identify and characterize this enzyme, a combined proteomic and transcriptomic approach was applied. To evaluate the amount of 5-CQA as well as 3,5-diCQA chromatographic analyses were performed. In general, the manuscript is well organized. It includes methodology with sufficient details for verification. The results are clearly presented and adequately supported with tables and figures. According to the reviewer's opinion, this paper contains enough new material that is worth publishing.

Nevertheless, the current version of the paper is too much like a fragment of a doctoral dissertation than a manuscript of a publication. Therefore, the manuscript should be revised by eliminating too extensive fragments of basic book knowledge from the Introduction Section. The division of the Discussion Section into subsections will certainly facilitate the understanding of information flow and the obtained results discussion. The introduction of a short summary section will also sharpen the message and achievements of the authors. In addition, all personal phrases like 'we showed / we noticed / we observed' should be removed, in accordance with generally accepted principles of writing scientific papers .

I consider that this paper can be published in the Communications Biology after minor revisions.

Reviewer #3 (Remarks to the Author):

The paper entitled "A GDSL Lipase-like from *Ipomoea balatas*, a key player for the 3,5-diCQA synthesis, catalyzed efficient production of 3,5-diCQA when expressed in *Pichia pastoris*" studies the synthesis of 3,5-DiCQA by recombinant IbICS. The work also includes results on the identification and functional characterization of the IbHCT and the cloning, expression and characterization of the IbICS. The scientific quality of the work is high, especially in the molecular biology studies. However, there are some points related to the enzyme expression and reaction study that should be revised. The comments are listed below:

1. When authors carry out the functional characterization of the IbICS they compare the "level of production" between the *E.coli* and *S. cerevisiae* system by performing a western blot analysis. They claim that *E. coli* leads to higher "level of production". However, they do not specify if they have previously prepared the samples by normalizing the cell concentrations (or at least it has not been specified in the material and methods section). Aiming to compare two different expression systems based on yeast or bacteria they should be compared in terms of enzyme quantity/activity per dry cell weight (or CFU, or Wet cell weigh, etc...), if not, they are not really comparing the "level of production" of these systems.
2. When authors perform the production of 3,5-DiCQA adding different amounts of CQA to the culture medium of *P. pastoris* they claim that the bioconversion allowed a 7.5-fold increase of the product compared to the native extract. In this case, they have not tested if they are using the same amount of enzyme in both experiments. This values should be known to really compare both

reaction results.

3. In caption of figure 7 they refer to P1, P2, P3 and P4 but they are not shown in the figure.

4. The manuscript is too long and sometimes is difficult to follow the results and discussion sections.

Answer to reviewers Miguel et al. Communications Biology

Reviewer #1

This manuscript reports a novel enzyme, IbICS, responsible for the production of dicaffeoylquinic acid in *I. balatas*. The authors also demonstrated a biotechnological production of the compound using pure monocaffeoylquinic acid and green coffee bean extract. Researchers have been long trying to identify this enzyme but have not succeeded. Therefore, the discovery of this enzyme is a breakthrough in natural product biosynthesis which is worth to publish. Although gene knockout experiment has not been performed, it is difficult to conclude as the author already suggested that IbICS is the main enzyme responsible for the biosynthesis. Additional experiments can be performed to support the finding. Please find my comments below.

Although English in the manuscript is understandable, it can be further improved.

Reviewer's comment:

Some parts of results are mixed with discussion. Please check, e.g. line 302-303 and 389-390.

Author's answer

Thank you for the suggestion. The text has been modified regarding the mix in line 302 and 303. Regarding lines 389-390, we don't understand the comment. This part is only a discussion based on results from other studies. In anyway, and in order to improve the flow, the whole manuscript has been edited again.

Reviewer's comment:

Line 63-64, please provide a reference for daily intake of CQA and diCQA.

Author's answer

Thank you for asking clarification as it is an important point. Three new references were added in the manuscript.

Reviewer's comment:

Figure 1, in the pathway leading to 3,5-DiCQA, quinate should be represented as one of the products catalyzed by IbICS.

Author's answer

We thank you for this comment. We have added quinic acid to the Figure 1.

Reviewer's comment:

Since HCT and HQT from some plants have been reported to be (partially) responsible for diCQA production, this manuscript only focused on HCT but did not raise the possibility of IbHQT in the diCQA production. I would like to see a discussion on why HQT is not mentioned/focused.

Author's answer

The reviewer is right. The work on IbHQT has been done in parallel. However, the enzyme exhibited a very weak HQT activity in the same reaction conditions. Therefore, we focused our research on IbHCT. In order to answer to your request, we add a sentence related to these data in the manuscript

Reviewer's comment:

Also, in term of gene expression analysis, I did not see any results on IbHCT and IbICS. Is the expression pattern of IbICS correlated with diCQA accumulation in different parts/tissues of sweet potato? How were the expressions of those genes under nitrogen depletion experiments?

Author's answer

The main objective of this study was to identify an enzyme able to produce 3,5-diCQA and to set up an efficient system for producing this molecule.

The identification of the physiological function of the enzyme would require additional experiments such as knock out experiments or silencing approaches. We didn't realize such study here. This will be the subject of another paper. We didn't realize the study of the expression pattern of genes and the comparison with the production of diCQA which would provide results that might be further criticized since the specialized metabolism is a complex network with a dynamic redistribution of fluxes (i.e the increase of several molecules could be redirected to other pathways).

Reviewer's comment:

Please add more details on CQA profiles in sweet potato used in this study. Different genotypes can accumulate different profiles of CQA. Besides 3,5-diCQA as I understand it is the major diCQA in sweet potato, I. batatas also accumulate different types of diCQA. I wonder if the biosynthesis of other diCQAs is derived from different enzymes or isomerization via non-enzymatic reaction. This can be discussed. I also like to see the MS2 spectrum of the product to confirm that it is 3,5-diCQA. Depending on chromatographic condition, two diCQA isomers can be eluted at almost the same retention time.

Author's answer

The comment of Reviewer 1 is right. In preliminary studies, we were able to show that 3,4- and 4,5-diCQA are degradation products. Their spontaneous formation can be observed under alkaline conditions. For example, in the following figure, we incubated the same amount of 3,5-diCQA in different pH at 30°C overnight. Under these conditions, 3,4-diCQA and 4,5-diCQA were detected at alkaline pH. The 3 diCQA isomers are well separated. The product generated by IbHCT corresponds perfectly to 3,5-diCQA standard. This figure has been added as suppl. Figure 11 in the manuscript and we explained it in the manuscript.

Reviewer's comment:

I could not find M&M on IbHCT activity assay demonstrated in Fig 2A. Fig 2A(i), the authors mentioned that they detected increased caffeic acid content in both positive and control reactions that possibly derived from degradation of caffeoyl-CoA or 5-CQA. I wonder why the increased caffeoyl-CoA content can also be observed.

Author's answer

Thank you for this justified remark. The M&M part of this experiment has been implemented in the text. Regarding the caffeic acid peak: this molecule is not present in the substrate. Therefore, it appears (and not increases) when the substrate is incubated in presence of bacterial extracts (containing IbHCT or not). This was already described by other authors like [1,2] e

- [1] M. Kojima, T. Kondo, An Enzyme in Sweet-Potato Root Which Catalyzes the Conversion of Chlorogenic Acid, 3-Caffeoylquinic Acid, to Isochlorogenic Acid, 3,5-Dicaffeoylquinic Acid, *Agric. Biol. Chem.* 49 (1985) 2467–2469.
- [2] L.A. Lallemand, C. Zubieta, S.G. Lee, Y. Wang, S. Acajjaoui, J. Timmins, S. McSweeney, J.M. Jez, J.G. McCarthy, A.A. McCarthy, A Structural Basis for the Biosynthesis of the Major Chlorogenic Acids Found in Coffee, *Plant Physiol.* 160 (2012) 249–260. <https://doi.org/10.1104/pp.112.202051>.

Reviewer's comment:

Fig. 4, please use color that clearly show the different between specialized metabolism and development/morphogenesis. What does “?” represent? Should you use IbICS in the figure?

Author's answer

We modified this figure. GDSL enzymes that were described to be involved in the specialized metabolism were coloured in orange. The “?” was replaced by “not defined”. IbGDSL was replaced by IbICS to be consistent with the text.

Reviewer's comment:

Figure 5, the authors did not mention the amounts of proteins loaded into each lane. So how could the production level in different host be compared?

Author's answer

Thank you for this comment. The reviewer is right, we can't compare the production level in different host. Consequently, we have modified the text. We have replaced “The level of production of the protein was efficient for both systems although higher in the prokaryotic one (Figure 5.A and B).” by “The level of production of the protein was efficient for both systems (Figure 5.A and B).”

Reviewer's comment:

Could the reason for having no enzymatic activity be that the authors did not remove signal peptide when expression the recombinant protein in *E. coli*? Signal sequence is usually removed when expressing in prokaryotic system. And in *S. cerevisiae*, the recombinant protein could not be solubilised, therefore, no activity can be detected.

Author's answer

Actually, the signal peptide was removed for the expression in *E. coli*. Thank you for your comment. We have corrected the M&M section. Concerning yeast, we added the possibility of a relationship between a poor solubilization and the absence of activity in the discussion section as suggested.

Reviewer's comment:

Figure 5C, the authors did not show purity of the purified enzyme in SDS-PAGE, only western blot result was provided. So I do not know the purity of the enzyme when determining kinetic parameters.

Author's answer

Thank you for this relevant comment for which we agree completely. We changed the sentence “The presence, identity and purity of the recombinant protein was assessed by SDS-PAGE and by immunoblot analysis using an anti-His antibody (Figure 5.C).” by “The presence and identity of the recombinant protein was assessed by immunoblot analysis using an anti-His antibody (Figure 5.C).” We have added in the present document a figure showing the purity of GDSL, purified with His tag affinity column. Since the IbICS was not purified to homogeneity (only 50% of purity) we only maintain the K_m values in the text.

Relatif purity of two samples of IbICS purified from agroinfiltrated *N. benthamiana* leaves thanks to Ni-NTA column (GE healthcare)

* Area of each spot was determined thanks to ImageJ software.
 ** Total soluble proteins were quantified thanks to Bradford assays

Spots	Area*	%	S1	S2
1	574	7,07		
2	410	5,04		
3	410	5,04		
4	712	8,8		
5 (GDSL)	4059	50	0,205 mg/ml	0,125 mg/ml
6	954	11,7		
7	1000	12,3		
Total	8119	100	0,41** mg/ml (TSP)	0,25** mg/ml (TSP)

SDS-PAGE of IbICS purified from agroinfiltrated *N. benthamiana* leaves thanks to Ni-NTA column and stained by Silver Nitrate

Reviewer's comment:

Line 310, Galieni et al. did the experiment on lettuce? Not yarrow.

Author's answer

We have modified the manuscript in order to answer to this request

Reviewer's comment:

Have the RNAseq raw data been deposited?

Author's answer

For the first submission we only submitted the IbICS and IbHCT that are the targets of this study. Based on the request of Reviewer 1 we submitted the whole raw data of the database to genbank. The ID is PRJNA647243 and was added to the manuscript.

Reviewer #2

The reviewed manuscript deals with the identification of a new enzyme belonging to the GDSL lipase-like family that is involved in the final stage of transformation of 5-CQA into 3,5-diCQA and able to increase the transformation efficiency by over 60%, making the transformation process a valuable technological tool that can be easily transferred on an industrial scale. To identify and characterize this enzyme, a combined proteomic and transcriptomic approach was applied. To evaluate the amount of 5-CQA as well as 3,5-diCQA chromatographic analyses were performed.

In general, the manuscript is well organized. It includes methodology with sufficient details for verification. The results are clearly presented and adequately supported with tables and figures. According to the reviewer's opinion, this paper contains enough new material that is worth publishing.

Reviewer's comment:

Nevertheless, the current version of the paper is too much like a fragment of a doctoral dissertation than a manuscript of a publication. Therefore, the manuscript should be revised by eliminating too extensive fragments of basic book knowledge from the Introduction Section.

Author's answer

The Introduction part about state of the art has been simplified. The whole text has been extensively revised to improve the flow of the manuscript.

Reviewer's comment:

The division of the Discussion Section into subsections will certainly facilitate the understanding of information flow and the obtained results discussion. The introduction of a short summary section will also sharpen the message and achievements of the authors.

Author's answer

Thank you for this proposal. In this new version we added subsections with titles to clarify the message. A short paragraph was also added to introduce discussion part.

Reviewer's comment:

In addition, all personal phrases like 'we showed / we noticed / we observed' should be removed, in accordance with generally accepted principles of writing scientific papers.

Author's answer

Thank you for this recommendation. These expressions have been removed from manuscript.

Reviewer's comment:

I consider that this paper can be published in the Communications Biology after minor revisions.

Author's answer

We thank you for this judgment.

Reviewer #3

The paper entitled "A GDSL Lipase-like from *Ipomoea balatas*, a key player for the 3,5-diCQA synthesis, catalyzed efficient production of 3,5-diCQA when expressed in *Pichia pastoris*" studies the synthesis of 3,5-DiCQA by recombinant *IbICS*. The work also includes results on the identification and functional characterization of the *IbHCT* and the cloning, expression and characterization of the *IbICS*. The scientific quality of the work is high, especially in the molecular biology studies. However, there are some points related to the enzyme expression and reaction study that should be revised. The comments are listed below:

Reviewer's comment:

When authors carry out the functional characterization of the *IbICS* they compare the "level of production" between the *E.coli* and *S. cerevisiae* system by performing a western blot analysis. They claim that *E. coli* leads to higher "level of production". However, they do not specify if they have previously prepared the samples by normalizing the cell concentrations (or at least it has not been specified in the material and methods section). Aiming to compare two different expression systems based on yeast or bacteria they should be compared in terms of enzyme quantity/activity per dry cell weight (or CFU, or Wet cell weigh, etc...), if not, they are not really comparing the "level of production" of these systems.

Author's answer

Thank you for this comment. We agree with both reviewer 1 and 3 that it is difficult to compare the production level in different host based on our first manuscript. The systems are very different, the growth rates are different, the cultures are done at different temperatures, the induction systems are different. This information is not essential at all to the results described in this article. Consequently, we have modified the text. We have replaced the sentence "The level of production of the protein was efficient for both systems although higher in the prokaryotic one (Figure 5.A and B)." by "The level of production of the protein was efficient for both systems (Figure 5.A and B).".

Reviewer's comment:

When authors perform the production of 3,5-DiCQA adding different amounts of CQA to the culture medium of *P. pastoris* they claim that the bioconversion allowed a 7.5-fold increase of the product compared to the native extract. In this case, they have not tested if they are using the same amount of enzyme in both experiments. These values should be known to really compare both reaction results.

Author's answer

Thank you for this remark. Actually, we used the same enzyme batch for realizing this experiment. We therefore used the same amount of IbICS in each test.

Reviewer's comment:

In caption of figure 7 they refer to P1, P2, P3 and P4 but they are not shown in the figure.

Author's answer

The comment is right. The figure has been modified during time and the caption has not been changed. We removed this information.

Reviewer's comment:

The manuscript is too long and sometimes is difficult to follow the results and discussion sections.

Author's answer

Thank you for this comment. To clarify the message, subsections with titles were added in the results and discussion sections. The whole manuscript has been edited to shorten it and make it easier for readers.

REVIEWERS' COMMENTS:

Reviewer #1 (Remarks to the Author):

The authors have responded all of my comments and the manuscript has been improved. I only have very minor comments for correction.

Figure 1, The word "Shikimate" in the molecule of caffeoyl shikimate is incomplete.

Figure 2, information on P3 is missing and there are two (iii) in 2B. This must be corrected.